# Evaluation of Correlations between Genetic Variants and High-Resolution Computed Tomography Patterns in Idiopathic Pulmonary Fibrosis

**DOI:** 10.3390/diagnostics11050762

**Published:** 2021-04-23

**Authors:** Elisa Baratella, Barbara Ruaro, Fabiola Giudici, Barbara Wade, Mario Santagiuliana, Francesco Salton, Paola Confalonieri, Michele Simbolo, Aldo Scarpa, Saverio Tollot, Cristina Marrocchio, Maria Assunta Cova, Marco Confalonieri

**Affiliations:** 1Department of Radiology, Cattinara Hospital, University of Trieste, 34127 Trieste, Italy; savix.st@gmail.com (S.T.); cristinamarrocchio@gmail.com (C.M.); m.cova@fmc.units.it (M.A.C.); 2Department of Pulmonology, University Hospital of Cattinara, 34127 Trieste, Italy; barbara.ruaro@yahoo.it (B.R.); mario@marionline.it (M.S.); francesco.salton@gmail.com (F.S.); paola.confalonieri.24@gmail.com (P.C.); marco.confalonieri@asugi.sanita.fvg.it (M.C.); 3Biostatistics Unit, Department of Medicine, Surgery and Health Sciences, Cattinara Hospital, University of Trieste, 34127 Trieste, Italy; fgiudici@units.it; 4Unit of Biostatistics, Epidemiology and Public Health, Department of Cardiac, Thoracic, Vascular Sciences and Public Health, University of Padua, 35131 Padua, Italy; 5AOU City of Health and Science of Turin, Department of Science of Public Health and Pediatrics, University of Torino, 10126 Torino, Italy; barbarajenniferhellen.wade@unito.it; 6Section of Pathology, Department of Diagnostics and Public Health, University of Verona, 37219 Verona, Italy; simbolo.michele@hotmail.it (M.S.); aldo.scarpa@univr.it (A.S.)

**Keywords:** interstitial lung disease, idiopathic pulmonary fibrosis, high-resolution computed tomography (HRCT), familial idiopathic pulmonary fibrosis

## Abstract

Background. Idiopathic pulmonary fibrosis (IPF) is a progressive fibrosing interstitial lung disease (ILD). This prospective observational study aimed at the evaluation of any correlation between genetic variants associated with IPF susceptibility and high-resolution computed tomography (HRCT) patterns. It also aimed at evidencing any differences in the HRTC pattern between the familial and sporadic form at diagnosis and after two years. Methods. A total of 65 IPF patients (mean age at diagnosis 65 ± 10) were enrolled after having given written informed consent. HRCT and genetic evaluations were performed. Results. A total of 19 familial (mean age 62 ± 15) and 46 sporadic (mean age 70 ± 9) IPF patients were enrolled. A statistically significant difference was evidenced in the HRTC pattern at diagnosis between the two groups. Sporadic IPF patients had a predominantly usual interstitial pneumonia (UIP) pattern compared with those patients with familial IPF (60.0% vs. 21.1%, respectively). Moreover, familial IPF patients had more alternative diagnoses than those with sporadic IPF (31.6% vs. 2.2%, respectively). Furthermore, there was a slight increase in the typical UIP pattern in the familial IPF group at two years from diagnosis. Conclusions. Genetic factors play a pivotal role in the risk of developing IPF. However, further studies are required to clarify how these genetic factors may guide clinical treatment decisions.

## 1. Introduction

Idiopathic pulmonary fibrosis (IPF) is a progressive and almost invariably fatal interstitial lung disease (ILD), of unknown etiology, which is characterized by progressive fibrosis and loss of lung function [1,2]. Evidence suggests that the incidence of IPF is rising [1,2,3,4]. A recent analysis of a UK-based primary care database calculated a rise in incidence of 78% between 2000 and 2012, as well as a doubling of prevalence, estimated at 38.8 per 100,000, with a consequential growing economic burden on global health care [2,3]. Although historical data suggest a median time from IPF diagnosis to death of only 2–3 years, long-term survival is likely to increase as patients are diagnosed earlier and are given treatment able to slow down disease progression [1,2,3,4]. Prompt diagnosis of IPF is a must so as to allow for appropriate care and support with the administration of anti-fibrotic therapy and evaluation for lung transplantation [1,2,3]. Accurate diagnosis of IPF is of vital importance, as other forms of ILD have similar clinical presentations to IPF but necessitate different treatment strategies [4,5,6,7,8]. 

As the likelihood that specific radiologic features reflect IPF is not absolute but depends on the clinical context, the high-resolution computed tomography (HRCT) pattern may be inconclusive, and clinical information may further improve HRCT scan interpretation [9,10,11,12]. 

Several mutations have been associated with an increased risk of IPF, including those in genes for surfactant proteins (SFTPA2, SFTPC), telomerase reverse transcriptase (TERT), the RNA component of telomerase (TERC), and TOLLIP and MUC5B, all of which play important roles in lung host defense [13,14,15,16,17,18,19,20,21,22,23,24,25,26,27,28,29,30,31,32,33,34]. Emerging evidence supports the hypothesis that different polymorphisms may be associated with different patterns of fibrosis observed using HRCT [12,13,14,15,16,17,18,19,20,21,22,23,24,25,26,27,28,29,30,31,32,33,34].

This study aimed at evidencing any correlations between genetic variants associated with IPF susceptibility and HRCT patterns and at determining any differences between familial and sporadic IPF. 

## 2. Materials and Methods 

### 2.1. Patient Population

A total of 65 patients affected by familial and sporadic forms of IPF were enrolled in the study diagnosed according to the 2018 criteria [9] during routine clinical assessment in our pulmonology department and after obtaining written informed consent from January 2016 to November 2018. A complete medical history was collected, and all patients underwent a clinical examination. Demographic data, i.e., age and gender, were also recorded (see Table 1). The inclusion criterion was diagnosis of IPF (familial or sporadic form) according to the latest guidelines [3,4,9,10]. The familial form is defined as when there are two or more cases of the disease in the same family. 

The exclusion criteria were: subjects with insufficient clinical information (i.e., gender, age at diagnosis) and lack of two-year follow-up. Patients with inadequate HRCT examinations, i.e., HRCT, who could not be evaluated due to motion artifacts were also excluded. HRCT scanning (described in Section 2.2) and genetic analysis (described in Section 2.3) were performed for all patients.

### 2.2. High-Resolution Computed Tomography 

The imaging protocol was as follows: all patients had a volumetric HRCT scan, performed during a single breath hold at full inspiration in a supine position. If subtle ground-glass opacities or reticulations were not prominent and were located in the subpleural regions of the lung bases, an additional scan was acquired in the prone position to differentiate these findings due to gravitational phenomena from ILD [9,10,11,12]. 

Moreover, if prominent mosaic perfusion was evidenced at full inspiration images, then an additional scan was performed at the end of a forced expiration in the supine position to identify any air trapping.

The HRCT scans were performed with a 256-row multidetector CT system (Brilliance iCT 256, Philips, Best, The Netherlands), and the technical parameters were: rotation time, 270 ms; beam collimation, 128 × 2 × 0.625 mm; normalized pitch, 0.975; *z*-axis coverage, 160 mm; reconstruction interval, 0.3 mm; section reconstruction thickness, 1 mm; tube voltage, 120 kV; tube current (effective mA), 280–400 depending on patient size; and field of view, 40 cm. 

The images were analyzed with consensus by two senior experienced thoracic imaging radiologists (16 and 10 years of experience); any discrepancies in imaging interpretation were resolved by obtaining the opinion of a third radiologist with 20 years of experience. The CT images were analyzed at standard lung window settings (a window level of −600 HU and a window width of 1600 HU) on a picture archiving and communications system (PACS)-integrated workstation (19-inch TFT display, resolution 2560~1600 pixels) [9,10,11,12]. 

The first task of the radiologists was to classify the HRCT pattern into four categories according to the Fleischner Society classification of fibrotic lung diseases: Usual interstitial pneumonia (UIP): the presence of honeycombing in association with a reticular pattern with or without traction bronchiectasis/bronchiolectasis, with a subpleural and basal predominance;Probable UIP: the presence of a reticular pattern in association with traction bronchiectasis/bronchiolectasis without honeycombing and with a subpleural and basal predominance;Indeterminate UIP: the presence of subtle reticulation or ground-glass opacities, with a subpleural and basal predominance;Alternative diagnosis: the presence of findings suggestive of an alternative diagnosis included mosaic attenuation, predominant ground-glass opacities, consolidations or signs of fibrosis with an atypical distribution in the upper-mid lung [9,10,11].

Then, radiologists evaluated a follow-up HRCT, performed two years after the first HRCT scan, to assess the current radiological pattern and whether it had remained unchanged, was stable or had worsened.

### 2.3. Genetic Analysis 

DNA extraction and quantification of blood DNA was prepared by QIAamp Blood Mini kit (Qiagen, Germany). DNA quality was further evaluated by PCR analysis by the BIOMED 2 PCR multiplex protocol with PCR products analyzed by DNA 1000 Assay (Life Technologies, USA) on the Agilent 2100 Bioanalyzer on-chip electrophoresis (Agilent Technologies, USA), as previously described by Simbolo et al. [35]. 

Next-Generation Sequencing of Multiplex PCR Amplicons Ampliseq multigene custom panel was designed to explore all exons of ABCA3, DKC1, PARN, RTEL1, SFTPA2, SFTPC, TERC and the TERT genes. Two additional regions were added to investigate the putative promoter of MUC5B and TERT (chr11: 1241210-1241230 and chr5: 1295105-1297000). Twenty nanograms of DNA was used for multiplex PCR amplification, followed by ligation of a specific barcode sequence to each sample for identification. Emulsion PCR was performed to construct the libraries of clonal sequences using the Ion 27 OneTouchTM OT2 System (Life Technologies). The quality of the libraries obtained was evaluated by the Agilent 2100 Bioanalyzer on-chip electrophoresis (Agilent Technologies). Sequencing was run on the Ion Torrent Personal Genome Machine (PGM, Life Technologies) loaded with 318v2 chips. Data analysis, including alignment to the hg19 human reference genome and variant calling, was carried out by the Torrent Suite Software v.5.0 (Life Technologies). Filtered variants were annotated by a custom pipeline, based on vcflib (https://www.github.com/ekg/vcflib accessed on 31 Decembre 2018), SnpSift [36] the Variant Effect Predictor (VEP) software [37] and the NCBI RefSeq database.

### 2.4. Statistical Analysis

Statistical analysis on the familial and sporadic form of ILD and different genetic mutations was performed by statistical analysis software R (version 4.0.2, 2020). Descriptive statistics of patient demographics and clinical characteristics were reported as frequencies (proportions) for categorical variables and mean (standard deviation) (SD) for continuous variables. The two groups (familial and sporadic IPF) were compared for age by the Student’s t-test, whilst the Chi-Squared test or Fischer Exact test was performed to identify differences in categorical variables (smoking status, HRCT patterns and any worsening of the pattern at two years). A *p*-value of <0.05 was considered statistically significant.

## 3. Results

The study population included 65 patients affected by IPF (mean age at diagnosis 65 ± 10), 19/65 the familial form (mean age at diagnosis 62 ± 15) and 46/65 the sporadic form (mean age at diagnosis 70 ± 9) (see Table 1).

The analysis of the clinical characteristics of the two populations showed that:(a)the age at onset of the disease was lower in patients with the familial form than in those with the sporadic form;(b)there was no significant difference in gender for the familial form, whilst there was a prevalence of males in the sporadic form;(c)moreover, no current smokers were observed in the group of patients affected by the familial form, whilst there were numerous current smokers in the sporadic form group.

At diagnosis, there was a statistically significant difference in the HRTC pattern between the two groups: sporadic IPF patients had a predominantly typical UIP pattern compared to familial IPF patients (60.0% vs. 21.1%, respectively). There was a predominance of an alternative diagnosis HRTC pattern in familial IPF compared to sporadic IPF (31.6% vs. 2.2%, respectively) (see Figure 1, Figure 2 and Figure 3). The HRCT for 6/19 patients with a familial pulmonary fibrosis at the onset of symptoms had an alternative diagnosis pattern: this was due to the presence of a fibrotic nonspecific interstitial pneumonia (NSIP) pattern in 4/19 patients, whilst in 2/19 patients it was due to the presence of honeycombing with upper-mid lung distribution (see Figure 1). At the onset of symptoms, only 1/46 sporadic IPF patients had an alternative diagnosis HRCT pattern, due to the presence of honeycombing with upper-mid lung prevalence and mosaic attenuation; the definitive diagnosis of IPF was made by biopsy (see Figure 2 and Figure 4). After 2 years from diagnosis in the familial IPF group, we observed a slight increase in the typical UIP pattern at HRTC evaluation (see Figure 3 and Figure 5). 

Furthermore, when we evaluated the baseline HRTC patterns versus HRCT patterns at 2 years from diagnosis (paired comparisons of patients in two different time points), both groups had similar percentages of patients with a worsened HRTC evaluation (63.2% familial vs. 69.8% sporadic) (see Figure 2 and Figure 3).

IPF patients may have more than one mutation and the number of mutations per patient averages 2.7 (min: 1, max: 5). Rare mutations were observed in our study groups, as listed in Table 2 and Table 3. MUC5B rs35705950 mutation was present in a minority of our patients and associated with a non-typical UIP pattern at HRCT scan. Two-thirds of patients with a TERT mutation at baseline had a typical UIP pattern (66.6%) (see Figure 2 and Figure 4), whilst the UIP pattern was observed in 1/15 patients without this mutation (6.6%) (*p*-value = 0.033). Furthermore, patients with SFTPC mutation had a frequent, but not statistically significant, indeterminate or alterative pattern (see Table 4) (see Figure 5).

## 4. Discussion

To the best of our knowledge, this is the first study to evaluate both the possibility of a correlation between genetic variants associated with IPF susceptibility and high-resolution computed tomography (HRCT) patterns and any differences between familial and sporadic IPF on the basis of the HRCT with a long follow-up (2 years). 

Although the number of patients enrolled is small, our data do reflect those in international literature, i.e., patients with the familial form were younger at diagnosis, the sporadic form is more frequent in males and this form has a higher proportion of smokers [12,13,14,38,39,40,41,42,43]. During the follow-up, we observed a significant variation in the HRCT pattern among the IPF patients with genetic mutations. The prevalence of genetic mutations in our prospective population of IPF patients was 29.2%. This is in line with the updated literature indicating at least 25–35% of patients with IPF having specific genetic mutations [34,35,36,37,38,39,40,41,42,43]. In our study population, among the most common genetic mutations for IPF, the TERT mutation was mostly associated at the time of diagnosis with an HRCT typical UIP pattern (66.6%), whilst the other mutations were mostly associated with an indeterminate or alternative HRCT pattern according to the current international guidelines [9,10,11,12]. This is the case of MUC5B rs35705950 mutation, ABCA3 mutation and the mutation in surfactant proteins, which were associated in our study with a predominant atypical pattern of pulmonary fibrosis, but not the classical UIP pattern at the time of diagnosis. Several studies reported that the MUC5B promoter polymorphism was associated with asymptomatic ILD in the general population. Interestingly, this association was frequently observed in older persons, and it did not appear to be influenced by cigarette smoking [30,33]. 

After 2 years, the lung HRCT pattern of most patients with genetic mutations moved towards a more typical UIP pattern. These observations lead us to postulate that the etiology and pathogenesis of IPF could change in the early phases of the disease according to the presence of specific driver mutations, but it becomes more similar when pulmonary fibrosis progresses to a more advanced stage. Thus, an integrated approach to patients with IPF combining genetic variants and HRCT scan patterns could help to better identify different natural history and clinical evolution, suggesting a need for more targeted treatment to ameliorate disease prognosis and outcomes. At the time of diagnosis and during the follow-up, an HRCT scan, performed and interpreted by an expert radiologist, still plays a key role. However, the likelihood that specific radiologic features reflect IPF is dependent on the clinical context, and according to our data, probably also by the genetic variants. Although both radiologists and pulmonologists can identify findings suggestive of a fibrosing ILD, both need the input of one from the other to improve the differential diagnosis process. When there is doubt regarding the diagnosis, an integrated approach is required and a working diagnosis may be appropriate. However, as new information pertinent to the patient’s diagnosis becomes available over time, it should be reviewed at regular intervals. Furthermore, genetic tests for common or rare mutations should be offered to patients with a history of familial pulmonary fibrosis to stratify disease course and risk. The genetic evaluation could be also helpful for estimating the risk of IPF for patients’ family members [9,10,11,12,44]. High-risk groups should be screened with genetic tests and by HRCT to prompt recognize early signs of ILD. 

Updated diagnostic guidelines for IPF will hopefully reflect the growing knowledge of genetic components of the disease in a way that ameliorates the clinical course and the therapeutic approach of these patients. 

## 5. Conclusions

In this study, we described the association between HRCT patterns and genetic variants in IPF. We confirmed the central role of HRCT scans in the diagnosis of IPF, and we demonstrated that HRTC evaluation may be enriched by genetic information to ameliorate the diagnosis, the follow-up and the therapeutic approach in this complex disease.

## Figures and Tables

**Figure 1 diagnostics-11-00762-f001:**
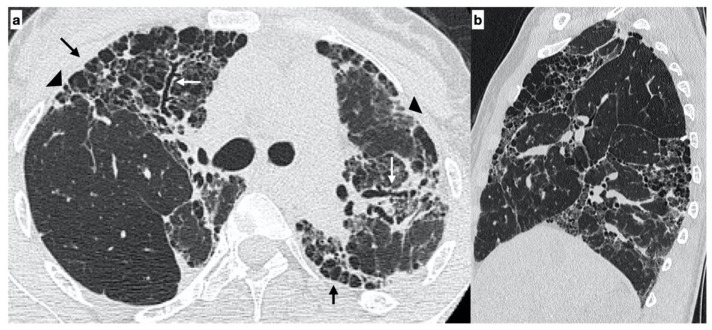
A 29-year-old female with an “alternative diagnosis pattern” on HRCT at onset of respiratory symptoms. The axial HRCT image shows the presence of honeycombing (black arrows), reticulations (arrowheads) and traction bronchiectasis/bronchiolectasis (white arrows) with a prevalent upper-mid lung distribution (**a**). The sagittal plane better evidences the atypical distribution of fibrotic alterations (**b**). Genetic analysis demonstrated the following mutations: ABCA3, SFTPA2, promoter-TERT, SFTPC.

**Figure 2 diagnostics-11-00762-f002:**
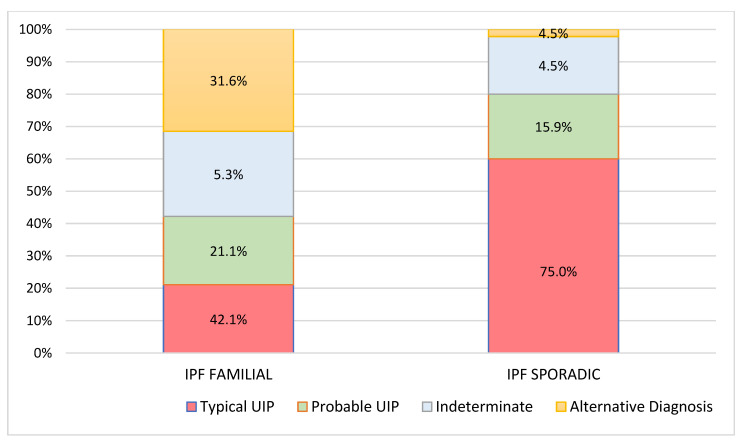
Baseline High-resolution computed tomography (HRCT) patterns with respect to Familial and Sporadic Idiopathic pulmonary fibrosis (IPF).

**Figure 3 diagnostics-11-00762-f003:**
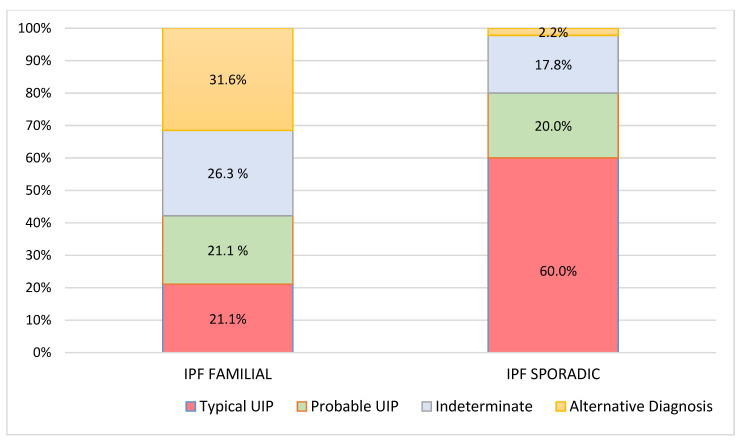
High-resolution computed tomography (HRCT) patterns at 2 years of follow-up respect to familial and sporadic idiopathic pulmonary fibrosis (IPF).

**Figure 4 diagnostics-11-00762-f004:**
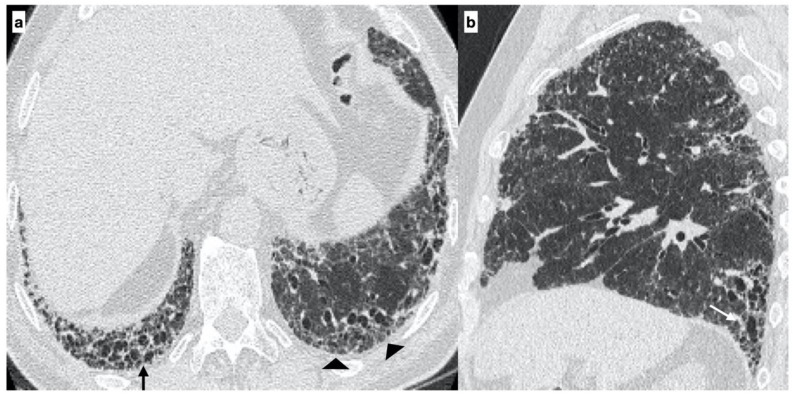
A 67-year-old male with “UIP pattern” on HRCT at the onset of respiratory symptoms. Axial HRCT plane shows honeycombing (black arrows) and reticulations (arrowheads) with a basal predominant distribution (**a**); traction bronchiectasis (white arrows) and distribution of alterations are better visible on the sagittal plane (**b**). Genetic analysis demonstrated the following types of mutations: SFTPA2, promoter-TERT, TERT.

**Figure 5 diagnostics-11-00762-f005:**
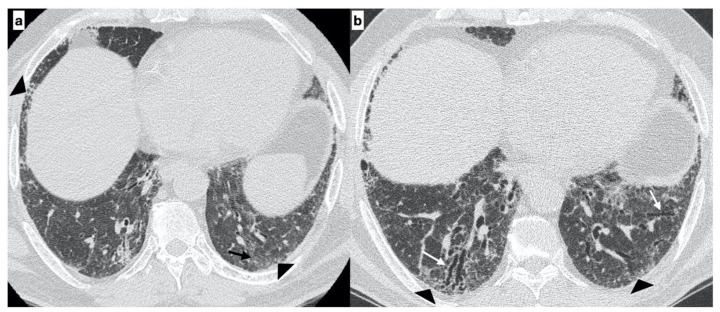
A 64-year-old male with an “Indeterminate pattern” on HRCT at onset of respiratory symptoms. The baseline axial HRCT image evidences subtle ground-glass opacities (black arrows) and reticulation (arrowheads) located in the subpleural regions of the lung bases with a predilection for the peripheral areas of both lungs (**a**). Follow-up HRCT, performed two years later, evidenced the worsening of the HRCT pattern due to the presence of extensive reticulations (arrowheads) and traction bronchiectasis/bronchiolectasis (white arrows) with a basal predominance (probable UIP pattern) (**b**). Genetic analysis demonstrated the following mutations: SFTPA2, promoter-TERT, SFTPC.

**Table 1 diagnostics-11-00762-t001:** Characteristics of the study population at baseline.

Variables	Familial (*n* = 19)	Sporadic (*n* = 46)	*p*-Value
Age of Diagnosis (mean ± SD)	62 ± 15	70 ± 9	0.012
Gender (N, %)			
Female	9 (47.4%)	13 (28.3%)	0.159
Male	10 (52.6%)	33 (71.7%)	
Smoking Status (N, %)			
Non-smoker	7 (41.2%)	11 (23.9%)	
Former smoker	9 (52.9%)	16 (34.8%)	0.027
Smoker	1 (5.9%)	19 (41.3%)	
Not available	2	0	
Pattern at Diagnosis (N, %)			
typical UIP	4 (21.1%)	27 (60.0%)	
probable UIP	4 (21.1%)	9 (20.0%)	0.001
Indeterminate	5 (26.3%)	8 (17.8%)	
Alternative Diagnosis	6 (31.6%)	1 (2.2%)	
Pattern at 2 years (N, %)			
Typical UIP	8 (42.1%)	33 (75.0%)	0.014
Probable UIP	4 (21.1%)	7 (15.9%)	
Indeterminate	1 (5.3%)	2 (4.5%)	
Alternative Diagnosis	6 (31.6%)	2 (4.5%)	
Worsening of the pattern after two years (N, %)			
Yes	12 (63.2%)	30 (69.8%)	0.769
Invariate	7 (36.8%)	13 (30.2%)	

**Table 2 diagnostics-11-00762-t002:** Frequency of mutations.

Gene	N of Mutation	%
PROMOTER-TERT	15	83.3%
SFTPA2	12	66.7%
SFTPC	8	44.4%
PROMOTER-MUC5B	6	33.3%
TERT	3	16.7%
TERC	3	16.7%
ABCA3	1	5.6%
DKC1	0	0.0%
RTEL1	0	0.0%

**Table 3 diagnostics-11-00762-t003:** Frequency and type of alleles polymorphism.

Mutation	Polymorphism	N	%
SFTPA2	rs1965708 (Q223K)	4	33.3%
	rs17886395 (A91P)	3	25.0%
	rs1965708 (Q223K); rs17886395 (A91P)	5	41.7%
ABCA3	Gly964Asp	1	100%
Promoter-MUC5B	rs35705950	1	100%
Promoter-TERT	rs143938607	1	6.7%
Promoter-TERT	rs2735940	4	26.7%
	rs2735940; rs2736109	6	40.0%
	rs2735940; rs3215401; rs2736109	2	13.3%
	rs2736109	1	6.7%
	rs33977403	1	6.7%
SFTPC	rs4715 (T138N); rs1124 (S186N)	2	25.0%
	rs4715 (T138N)	6	75.0%
TERT	rs35719940 (A1062T)	1	16.7%
	R756C	2	33.3%
TERC	rs2293607	3	50.0%

**Table 4 diagnostics-11-00762-t004:** Correlation between mutations and HRTC pattern at time of diagnosis.

Variables	Typical UIP	Probable UIP	Indeterminate	Alternative Diagnosis	*p*-Value
Promoter-TERT					
Yes	2	3	5	5	*p* = 0.651
No	1	1	0	1	
SFTPA2					
Yes	2	3	4	3	*p* = 0.830
No	1	1	1	3	
SFTPC					*p* = 0.168
Yes	0	1	4	3	
No	3	3	1	3	
Promoter-MUC5B					
Yes	1	0	3	2	*p* = 0.364
No	2	4	2	4	
TERT					
Yes	2	1	0	0	*p* = 0.033
No	1	3	5	6	
TERC					
Yes	2	0	1	1	*p* = 0.853
No	1	4	4	5	

## Data Availability

The data presented in this study are available on request from the corresponding author.

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
