# Peer review of "Evaluation of Correlations between Genetic Variants and High-Resolution Computed Tomography Patterns in Idiopathic Pulmonary Fibrosis"

_diagnostics, 2021, doi:10.3390/diagnostics11050762_

Round 1
Reviewer 1 Report
Baratella and Ruaro et al. have performed a neat association study between genetic variants and clinical profiling of idiopathic pulmonary fibrosis (IPF) obtained using high-resolution computed tomography (HRCT). Apart from demonstrating that HRCT can aide interpreting surgical lung biopsies from early and advanced stage IPF patients, their work points to several key genetic markers that could be harnessed in routine clinical practice to better diagnose IPF. The manuscript is well written and the data are presented in a compact and comprehensible format. Several formal points requiring further attention are summarized below.
1) Please change "Evaluation of any correlations between genetic variants and High-Resolution Computed Tomography patterns in Idiopathic Pulmonary Fibrosis" to either "Evaluation of correlations between genetic variants and high-resolution computed tomography patterns in idiopathic pulmonary fibrosis" or "Evaluation of Correlations between Genetic Variants and High-Resolution Computed Tomography Patterns in Idiopathic Pulmonary Fibrosis" (line 2).
2) Please format "elisa.baratella@gmail.com" consistent with the rest of the text (line 19).
3) Please format text on lines 20–26 consistent with the main text.
4) Please replace "form, at" with "form at" (line 24).
5) The authors claim that the average age at diagnosis of patients was 65±10 years in the Abstract section (line 25), however this fact is missing from the Results section.
6) Please change "65±10" to "65 ± 10" (line 25).
7) Please replace "62±15" with "62 ± 15 (line 26).
8) Please change "70±9" with "70 ± 9" (line 27).
9) Please change "IPF" to "IPF patients" (lines 27, 177).
10) "Sporadic IPF patients had a predominantly usual interstitial pneumonia (UIP) pattern than those with familial IPF patients" (line 28) seems not to be grammatically correct. Please rephrase.
11) It is not immediately clear what do the authors mean by "alternative diagnosis" in "Moreover, familial IPF patients had more alternative diagnosis than those with sporadic IPF" (line 30)?
12) Please replace "clinical decisions as to treatment" with "clinical treatment decisions" (line 35).
13) Please change "Interstitial lung disease · Idiopathic Pulmonary Fibrosis · High-resolution computed 36 tomography (HRCT) · Familial Idiopathic Pulmonary Fibrosis" (line 36) either to "interstitial lung disease, idiopathic pulmonary fibrosis, high-resolution computed tomography (HRCT), familial idiopathic pulmonary fibrosis" or "Interstitial Lung Disease, Idiopathic Pulmonary Fibrosis, High-resolution Computed Tomography (HRCT), Familial Idiopathic Pulmonary Fibrosis".
14) Please change "origin aetiology and" to "aetiology or origin, which" (line 41).
15) Please replace "progression" with "disease progression" (line 48).
16) "High-resolution computed tomography (HRCT) plays a central role in IPF diagnosis" (line 52) does not fit well to the preceding paragraph. Please fix.
17) Please change "context the" to "context, the" (line 55).
18) "and clinical information improves HRCT scan interpretation" (line 55) does not fit well the preceding text. Please revise.
19) "improves" may better read as "may improve" or "may further improve" (line 56).
20) "imaging findings" repeats twice in "Radiologist have the key task of understanding the imaging findings and patterns associated with IPF and other fibrotic lung diseases using their skill/experience to avoid imaging classification pitfalls and confounding imaging findings with other fibrotic lung diseases" (line 56).
21) "confounding imaging findings with other fibrotic lung diseases" (line 59) is not grammatically correct. Please rephrase.
22) "Noteworthy is the fact that" seems to be redundant in "Noteworthy is the fact that emerging evidence supports the hypothesis that different polymorphisms may be associated with different patterns of fibrosis on HRCT" (line 63).
23) Please replace "on" with something like "seen with" or "observed using" (line 66).
24) Please shorten "high-resolution computed tomography (HRCT)" to "HRCT" (line 68).
25) Please change "to determine" to "at determining" (line 69).
26) Please change "according to 2018 criteria (9) were enrolled, after obtaining written informed consent, during routine clinical assessment in our Pulmonology Department" to "were enrolled to the study diagnosed according to the 2018 criteria (9) during routine clinical assessment in our pulmonology department and after obtaining written informed consent" (line 73).
27) Please replace "Idiopathic Pulmonary Fibrosis" with "IPF" (line 78).
28) Please change "Familial(n=19)" to "Familial (n=19)" 2x, "62±15)" to "62 ± 15", and "70± (9)" to "70 ± 9" in Table 1.
29) Please move "Pattern at Diagnosis (N, %)" to the following page in Table 1.
30) Please replace "typical" with "Typical" 2x in Tables 1 and 4, Graphs 1 and 2.
31) Please change "probable" to "Probable" 2x in Table 1 and 4, Graphs 1 and 2.
32) Please replace "follow" with "follows" (line 91).
33) Please shorten "interstitial lung disease" to "ILD" (line 95).
34) Please change "position, to" to "position to" (line 99).
35) Please change "categories, according" to "categories according" (line 113).
36) Please replace "- Usual" with "Usual" (line 115).
37) Please change "-Probable" to "Probable" (line 118).
38) Please replace "-Indeterminate" with "Indeterminate" (line 121).
39) Please change "-Alternative" to "Alternative" (line 123).
40) Do the authors actually mean "HRCT scan" instead of "CT scan" (lines 127, 205, 259, 266)?
41) Please replace "quantification Blood" with "quantification of blood" (line 131).
42) Please change "Twenty nanograms" to "20 ng" (line 140).
43) Please replace "sequences, using" with "sequences using" (line 143).
44) Please incorporate "(Cingolani front genetic 2012, 136)" and "(McLaren 150 Bioninformatics 2010, 37)" as standard citations (line 150). Also, please note spelling mistake in the word "Bioninformatics".
45) Please change "Statistical Analysis" to "statistical analysis" (line 155).
46) Please change "(Standard Deviation-SD)" to "(standard deviation) (SD)" (line 157).
47) Please replace "Smoking Status" with "smoking status" (line 160).
48) Please change "(average age at diagnosis 62±15 )" to "(average age at diagnosis 62 ± 15)" (line 166).
49) Please replace "(average age at diagnosis 70±9)" with "(average age at diagnosis 70 ± 9)" (line 166).
50) Please change "table 1" to "Table 1" (line 167).
51) "sporadic IPF had a predominantly typical UIP pattern, than those with familial IPF patients" (line 177) is not grammatically correct. Please fix.
52) Please change "pattern, than" to "pattern than" (line 177).
53) Please define abbreviation for "NSIP" (line 182).
54) From "There was a slight increase in the typical UIP pattern in the familial IPF group, at 2 years from diagnosis" (line 186) is not clear to what condition is the increase in the typical UIP pattern in the familial IPF group being compared to? Please revise.
55) Please replace "group, at" with "group at" (line 187).
56) Please annotate HRCT images in Figures 1, 2, and 3 with respect to the pathofysiological features of IPF (honeycombing, reticulations, traction bronchiectasis/bronchiolectasis, fibrotic alterations, ground-glass opacities and reticulation, predilection, UIP pattern).
57) Please adjust the image height in Figures 1 and 3 so that images have equal size within the same figure.
58) Please add labels "a" and "b" to images in Figures 1, 2, and 3.
59) Please replace "Alternative Diagnosis" with "alternative diagnosis" (line 190).
60) Please change "The sagittal plane evidences the atypical distribution of fibrotic alterations better(b)" to "The sagittal plane better evidences the atypical distribution of fibrotic alterations (b)" (line 192).
61) Please replace "TERT-promoter-" to "promoter-TERT" (line 193).
62) Please remove all decimal places from the y axes in Graphs 1 and 2.
63) Please use decimal point instead of comma for values plotted in Graphs 1 and 2.
64) Please round numbers to only one decimal point for values plotted in Graph 1.
65) Please change "pattern ." to "pattern." (line 196).
66) The values of 63.2% for familial and 69.8% for sporadic HRTC evaluation in "Furthermore, both groups had similar percentages of patients with a worsened HRTC evaluation (63.2% familial vs 69.8% sporadic), at two years from diagnosis" (line 199) are not evident from Graph 2. Please fix.
67) Please replace "(63.2% familial vs 69.8% sporadic), at" with "(63.2% familial vs 69.8% sporadic) at" (line 200).
68) Please change "min: 1 max: 5" to "min: 1, max: 5" (line 203).
69) Please replace "Two/thirds" with "Two thirds" (line 205).
70) Please change "mutation, at" with "mutation at" (line 206).
71) Please replace "Typical" with "typical" (line 206).
72) Please change "on sagittal plane traction bronchiectasis and distribution of alterations are better visible" to "traction bronchiectasis and distribution of alterations are better visible on sagittal plane" (line 224).
73) Please replace "TERT -promoter-" with "promoter-TERT" (line 226).
74) Please change "TERT- promoter" to "promoter-TERT" (line 236).
75) Please replace "follow-up we" with "follow-up, we" (line 246).
76) Please change "indicate" with "indicating" (line 249).
77) "study population" could be shortened just to "study" (lines 250, 255).
78) Do the authors actually mean "IPF" instead of "PF" (lines 251, 258)?
79) Please define abbreviation for "pulmonary fibrosis" at the site of the first occurrence of this term (line 251).
80) Please replace "HRCT pattern indeterminate or alternative" with "indeterminate or alternative HRCT pattern" (line 253).
81) Please change "surfactant proteins mutation" to "mutation in surfactant proteins" (line 255).
82) What do the authors actually mean by "survey of Framingham" in "It is to consider that also in the large cohort survey of Framingham MUC5B rs35705950 and other familiar PF mutations were associated with asymptomatic interstitial lung abnormalities detected on CT scan among older people without an overt clinical picture of IPF" (line 257)?
83) Please replace "move" with "moved" or "had moved" (line 261).
84) "genetic" could read better as "driver" (line 264).
85) Please change "more" to "need for more" (line 267).
86) "prognosis" could be changed to "disease prognosis" (line 268).
87) Please replace "At time" with "At the time" (line 268).
88) Please change "follow-up a" to "follow-up, a" (line 269).
89) Please replace "play" with "plays" (line 269).
90) Please change "data probably" to "data, probably" (line 271).
91) "to add genetic information" sounds redundant in "Although both radiologists and pulmonologists can identify findings suggestive of a fibrosing ILD, both need input one from the other to add genetic information to improve the differential diagnosis process" (line 272).
92) "When there is doubt as to the diagnosis on the basis of the patient’s radiological and clinical features are required an integrated approach and a working diagnosis may be appropriate for some patients" (line 274) is not grammatically correct with respect to "as to the diagnosis on the basis of the patient’s radiological and clinical features are required".
93) "information pertinent to the patient’s diagnosis may become" could be shortened to "patient’s diagnostic information becomes" (line 276).
94) "High risk group should be carefully evaluated clinical and radiological and should be screened with genetic test and by HRCT to prompt recognize early sign of ILD" (line 281) is not grammatically correct with respect to "evaluated clinical and radiological and should be screened".
95) "Updated diagnostic guidelines for IPF will hopefully reflect the growing knowledge on genetic component of the disease in a way that the clinical course of patients with specific clinical and radiological features and be able to provide innovative information on the molecular signatures predictive of UIP" (line 284) is too long and is not grammatically correct with respect to "clinical and radiological features and be able to". Please revise.
96) The meaning of "Thus, that accounts for inherited factors and HRCT patterns could be associated with different therapeutic approach" (line 287) is vague and the sentence is not grammatically correct with respect to "that accounts for inherited factors and HRCT patterns could be". Please rephrase this concluding statement to better highlight the role of genetic diagnosis in informing IPF therapy.
97) Please change "we described some association between" to "we described the association between" or "we have established a link between" (line 291).
98) Please replace "at time" with "at the time" (line 292).
99) "HRCT scans play a key role in IPF diagnosis and how clinical information, provided by pulmonologists and other clinicians, can aid radiologists in the interpretation of HRCT scans, guiding the need for surgical lung biopsy and determining available pharmacologic therapies" (line 293) is not grammatically correct with respect to "and how clinical information, provided by pulmonologists and other clinicians, can aid radiologists in the interpretation of HRCT scans".
100) "The central role of HRCT scans in the diagnosis of IPF may be enriched by genetic information to provide a differentiated therapeutic approach according to the molecular characteristics" (line 296) does not provide a straightforward and elegant conclusion. Please rephrase. In addition it is not clear what molecular characteristics are the authors referring to? Also, note that the word "role" was already used in the preceding sentence "HRCT scans play a key role in IPF diagnosis and how clinical information, provided by pulmonologists and other clinicians, can aid radiologists in the interpretation of HRCT scans, guiding the need for surgical lung biopsy and determining available pharmacologic therapies" (line 293).
101) Please distinguish the contribution of "Mario Santagiuliana" from that of "Michele Simbolo" in the Author Contributions section.
102) Please format "published version of the manuscript" (line 302) consistent with the rest of the text.
103) Please change "funding" to "funding." (line 303).
104) Please replace "Helsinki" with "Helsinki." (line 305).
105) Please change "interest" to "interest." (line 306).
Author Response
Baratella and Ruaro et al. have performed a neat association study between genetic variants and clinical profiling of idiopathic pulmonary fibrosis (IPF) obtained using high-resolution computed tomography (HRCT). Apart from demonstrating that HRCT can aide interpreting surgical lung biopsies from early and advanced stage IPF patients, their work points to several key genetic markers that could be harnessed in routine clinical practice to better diagnose IPF. The manuscript is well written and the data are presented in a compact and comprehensible format. Several formal points requiring further attention are summarized below.
1) Please change "Evaluation of any correlations between genetic variants and High-Resolution Computed Tomography patterns in Idiopathic Pulmonary Fibrosis" to either "Evaluation of correlations between genetic variants and high-resolution computed tomography patterns in idiopathic pulmonary fibrosis" or "Evaluation of Correlations between Genetic Variants and High-Resolution Computed Tomography Patterns in Idiopathic Pulmonary Fibrosis" (line 2).
R: We thank the reviewer for this comment , which allows us to enhance the quality of the manuscript. As requested the title has been changed.
2) Please format "elisa.baratella@gmail.com" consistent with the rest of the text (line 19).
R: As requested, the lines have been reformatted.
3) Please format text on lines 20–26 consistent with the main text.
As requested, the lines have been reformatted.
4) Please replace "form, at" with "form at" (line 24).
R: As requested, the words have been replace
5) The authors claim that the average age at diagnosis of patients was 65±10 years in the Abstract section (line 25), however this fact is missing from the Results section.
R: As requested, the data have been added in the Results section.
6) Please change "65±10" to "65 ± 10" (line 25).
R: As requested the data have been changed.
7) Please replace "62±15" with "62 ± 15 (line 26).
R: As requested the data have been changed.
8) Please change "70±9" with "70 ± 9" (line 27).
R: As requested the data have been changed.
9) Please change "IPF" to "IPF patients" (lines 27, 177).
R: As requested the word “patients” has been added.
10) "Sporadic IPF patients had a predominantly usual interstitial pneumonia (UIP) pattern than those with familial IPF patients" (line 28) seems not to be grammatically correct. Please rephrase.
R: As requested, the words have been replace
11) It is not immediately clear what do the authors mean by "alternative diagnosis" in "Moreover, familial IPF patients had more alternative diagnosis than those with sporadic IPF" (line 30)?
R: As requested, the lines are rewritten.
12) Please replace "clinical decisions as to treatment" with "clinical treatment decisions" (line 35).
R: As requested the sentence has been changed.
13) Please change "Interstitial lung disease · Idiopathic Pulmonary Fibrosis · High-resolution computed 36 tomography (HRCT) · Familial Idiopathic Pulmonary Fibrosis" (line 36) either to "interstitial lung disease, idiopathic pulmonary fibrosis, high-resolution computed tomography (HRCT), familial idiopathic pulmonary fibrosis" or "Interstitial Lung Disease, Idiopathic Pulmonary Fibrosis, High-resolution Computed Tomography (HRCT), Familial Idiopathic Pulmonary Fibrosis".
R: As requested the sentence has been changed.
14) Please change "origin aetiology and" to "aetiology or origin, which" (line 41).
R: As requested the sentence has been changed.
15) Please replace "progression" with "disease progression" (line 48).
R: As requested the sentence has been changed.
16) "High-resolution computed tomography (HRCT) plays a central role in IPF diagnosis" (line 52) does not fit well to the preceding paragraph. Please fix.
R: As requested the sentence has been changed.
17) Please change "context the" to "context, the" (line 55).
R: As requested the sentence has been changed.
18) "and clinical information improves HRCT scan interpretation" (line 55) does not fit well the preceding text. Please revise.
R: As requested the sentence has been changed.
19) "improves" may better read as "may improve" or "may further improve" (line 56).
R: As requested the sentence has been changed.
20) "imaging findings" repeats twice in "Radiologist have the key task of understanding the imaging findings and patterns associated with IPF and other fibrotic lung diseases using their skill/experience to avoid imaging classification pitfalls and confounding imaging findings with other fibrotic lung diseases" (line 56).
R: As requested the sentence has been changed.
21) "confounding imaging findings with other fibrotic lung diseases" (line 59) is not grammatically correct. Please rephrase.
R: As requested the sentence has been changed.
22) "Noteworthy is the fact that" seems to be redundant in "Noteworthy is the fact that emerging evidence supports the hypothesis that different polymorphisms may be associated with different patterns of fibrosis on HRCT" (line 63).
R: As requested the sentence has been changed.
23) Please replace "on" with something like "seen with" or "observed using" (line 66).
R: As requested the sentence has been changed.
24) Please shorten "high-resolution computed tomography (HRCT)" to "HRCT" (line 68).
R: As requested the sentence has been changed.
25) Please change "to determine" to "at determining" (line 69).
R: As requested the sentence has been changed.
26) Please change "according to 2018 criteria (9) were enrolled, after obtaining written informed consent, during routine clinical assessment in our Pulmonology Department" to "were enrolled to the study diagnosed according to the 2018 criteria (9) during routine clinical assessment in our pulmonology department and after obtaining written informed consent" (line 73).
R: As requested the sentence has been changed.
27) Please replace "Idiopathic Pulmonary Fibrosis" with "IPF" (line 78).
R: As requested the acronym has been used.
28) Please change "Familial(n=19)" to "Familial (n=19)" 2x, "62±15)" to "62 ± 15", and "70± (9)" to "70 ± 9" in Table 1.
R: As requested, all the typos have been corrected.
29) Please move "Pattern at Diagnosis (N, %)" to the following page in Table 1.
R: As requested, all the typos have been corrected.
30) Please replace "typical" with "Typical" 2x in Tables 1 and 4, Graphs 1 and 2.
R: As requested, all the typos have been corrected.
31) Please change "probable" to "Probable" 2x in Table 1 and 4, Graphs 1 and 2.
R: As requested, all the typos have been corrected.
32) Please replace "follow" with "follows" (line 91).
R: As requested, the mistake has been corrected.
33) Please shorten "interstitial lung disease" to "ILD" (line 95).
R: As requested the acronym has been used.
34) Please change "position, to" to "position to" (line 99).
R: As requested, the comma has been removed.
35) Please change "categories, according" to "categories according" (line 113).
R: As requested, the comma has been removed.
36) Please replace "- Usual" with "Usual" (line 115).
R: As requested, the dash has been removed
37) Please change "-Probable" to "Probable" (line 118).
R: As requested, the dash has been removed
38) Please replace "-Indeterminate" with "Indeterminate" (line 121).
R: As requested, the dash has been removed
39) Please change "-Alternative" to "Alternative" (line 123).
R: As requested, the dash has been removed
40) Do the authors actually mean "HRCT scan" instead of "CT scan" (lines 127, 205, 259, 266)?
R: As requested, the mistake has been corrected.
41) Please replace "quantification Blood" with "quantification of blood" (line 131).
R: As requested, the mistake has been corrected.
42) Please change "Twenty nanograms" to "20 ng" (line 140).
R: As requested, all the typos have been corrected.
43) Please replace "sequences, using" with "sequences using" (line 143).
R: As requested, the comma has been removed.
44) Please incorporate "(Cingolani front genetic 2012, 136)" and "(McLaren 150 Bioninformatics 2010, 37)" as standard citations (line 150). Also, please note spelling mistake in the word "Bioninformatics".
R: As requested, all the mistakes have been corrected.
45) Please change "Statistical Analysis" to "statistical analysis" (line 155).
R: As requested, all the typos have been corrected.
46) Please change "(Standard Deviation-SD)" to "(standard deviation) (SD)" (line 157).
R: As requested, all the typos have been corrected.
47) Please replace "Smoking Status" with "smoking status" (line 160).
R: As requested, all the typos have been corrected.
48) Please change "(average age at diagnosis 62±15 )" to "(average age at diagnosis 62 ± 15)" (line 166).
R: As requested, all the typos have been corrected.
49) Please replace "(average age at diagnosis 70±9)" with "(average age at diagnosis 70 ± 9)" (line 166).
R: As requested, all the typos have been corrected.
50) Please change "table 1" to "Table 1" (line 167).
R: As requested, all the typos have been corrected.
51) "sporadic IPF had a predominantly typical UIP pattern, than those with familial IPF patients" (line 177) is not grammatically correct. Please fix.
R: As requested, the mistakes have been corrected.
52) Please change "pattern, than" to "pattern than" (line 177).
R: As requested, the comma has been removed.
53) Please define abbreviation for "NSIP" (line 182).
R: As requested, the abbreviation has been defined.
54) From "There was a slight increase in the typical UIP pattern in the familial IPF group, at 2 years from diagnosis" (line 186) is not clear to what condition is the increase in the typical UIP pattern in the familial IPF group being compared to? Please revise.
R: As requested, the lines are rewritten.
55) Please replace "group, at" with "group at" (line 187).
R: As requested, the comma has been removed.
56) Please annotate HRCT images in Figures 1, 2, and 3 with respect to the pathofysiological features of IPF (honeycombing, reticulations, traction bronchiectasis/bronchiolectasis, fibrotic alterations, ground-glass opacities and reticulation, predilection, UIP pattern).
R: As requested, the pathofysiological features of IPF are annotate.
57) Please adjust the image height in Figures 1 and 3 so that images have equal size within the same figure.
R: As requested, the images have equal size.
58) Please add labels "a" and "b" to images in Figures 1, 2, and 3.
R: As requested, the labels "a" and "b" to images are added.
59) Please replace "Alternative Diagnosis" with "alternative diagnosis" (line 190).
R: As requested, all the typos have been corrected.
60) Please change "The sagittal plane evidences the atypical distribution of fibrotic alterations better(b)" to "The sagittal plane better evidences the atypical distribution of fibrotic alterations (b)" (line 192).
R: As requested, all the typos have been corrected.
61) Please replace "TERT-promoter-" to "promoter-TERT" (line 193).
R: As requested, all the typos have been corrected.
62) Please remove all decimal places from the y axes in Graphs 1 and 2.
R: As requested, all the typos have been corrected.
63) Please use decimal point instead of comma for values plotted in Graphs 1 and 2.
R: As requested, all the typos have been corrected.
64) Please round numbers to only one decimal point for values plotted in Graph 1.
R: As requested, all the typos have been corrected.
65) Please change "pattern ." to "pattern." (line 196).
R: As requested, all the typos have been corrected.
66) The values of 63.2% for familial and 69.8% for sporadic HRTC evaluation in "Furthermore, both groups had similar percentages of patients with a worsened HRTC evaluation (63.2% familial vs 69.8% sporadic), at two years from diagnosis" (line 199) are not evident from Graph 2. Please fix.
R: We thank the reviewer for this clarification. In the Graph 2 (Figure2 in the revised version of the manuscript as suggested by reviewer 2) the percentages are referred to the HRCT patterns at 2 years of follow-up for both the two groups, INDEPENDENTLY from the basal HRCT patterns. Instead, percentages reported in the sentence "Furthermore, both groups had similar percentages of patients with a worsened HRTC evaluation (63.2% familial vs 69.8% sporadic), at two years from diagnosis" do not refer to Figure 2: they were calculated considering in each group the patients that from baseline to 2 years of follow-up were worsening (it is a paired comparison of HRCT patterns from two temporal point for each group). In the revised version of the manuscript we have modified the previous sentence: “Furthermore, when we evaluated baseline HRTC patterns versus HRCT patterns at 2 years from diagnosis (paired comparisons of patients in two different time points), both groups had similar percentages of patients with a worsened HRTC evaluation (63.2% familial vs 69.8% sporadic)”.
67) Please replace "(63.2% familial vs 69.8% sporadic), at" with "(63.2% familial vs 69.8% sporadic) at" (line 200).
R: As requested, all the typos have been corrected.
68) Please change "min: 1 max: 5" to "min: 1, max: 5" (line 203).
R: As requested, all the typos have been corrected.
69) Please replace "Two/thirds" with "Two thirds" (line 205).
R: As requested, all the typos have been corrected.
70) Please change "mutation, at" with "mutation at" (line 206).
R: As requested, all the typos have been corrected.
71) Please replace "Typical" with "typical" (line 206).
R: As requested, all the typos have been corrected.
72) Please change "on sagittal plane traction bronchiectasis and distribution of alterations are better visible" to "traction bronchiectasis and distribution of alterations are better visible on sagittal plane" (line 224).
R: As requested, all the mistakes have been corrected.
73) Please replace "TERT -promoter-" with "promoter-TERT" (line 226).
R: As requested, all the typos have been corrected.
74) Please change "TERT- promoter" to "promoter-TERT" (line 236).
R: As requested, all the typos have been corrected.
75) Please replace "follow-up we" with "follow-up, we" (line 246).
R: As requested, the comma has been added.
76) Please change "indicate" with "indicating" (line 249).
R: As requested, all the typos have been corrected.
77) "study population" could be shortened just to "study" (lines 250, 255).
R: As requested, all the typos have been corrected.
78) Do the authors actually mean "IPF" instead of "PF" (lines 251, 258)?
R: As requested, all the typos have been corrected.
79) Please define abbreviation for "pulmonary fibrosis" at the site of the first occurrence of this term (line 251).
R: As requested, all the typos have been corrected.
80) Please replace "HRCT pattern indeterminate or alternative" with "indeterminate or alternative HRCT
pattern" (line 253).
R: As requested, the lines are rewritten.
81) Please change "surfactant proteins mutation" to "mutation in surfactant proteins" (line 255).
R: As requested, the lines are rewritten.
82) What do the authors actually mean by "survey of Framingham" in "It is to consider that also in the large cohort survey of Framingham MUC5B rs35705950 and other familiar PF mutations were associated with asymptomatic interstitial lung abnormalities detected on CT scan among older people without an overt clinical picture of IPF" (line 257)?
R: As requested, the lines are rewritten.
83) Please replace "move" with "moved" or "had moved" (line 261).
R: As requested, the mistake has been corrected.
84) "genetic" could read better as "driver" (line 264).
R: As requested, the word has been corrected.
85) Please change "more" to "need for more" (line 267).
R: As requested, the lines are rewritten.
86) "prognosis" could be changed to "disease prognosis" (line 268).
R: As requested, the word has been added.
87) Please replace "At time" with "At the time" (line 268).
R: As requested, the word has been added.
88) Please change "follow-up a" to "follow-up, a" (line 269).
R: As requested, the comma has been added.
89) Please replace "play" with "plays" (line 269).
R: As requested, the mistake has been corrected.
90) Please change "data probably" to "data, probably" (line 271).
R: As requested, the comma has been added.
91) "to add genetic information" sounds redundant in "Although both radiologists and pulmonologists can identify findings suggestive of a fibrosing ILD, both need input one from the other to add genetic information to improve the differential diagnosis process" (line 272).
R: As requested, the lines are rewritten.
92) "When there is doubt as to the diagnosis on the basis of the patient’s radiological and clinical features are required an integrated approach and a working diagnosis may be appropriate for some patients" (line 274) is not grammatically correct with respect to "as to the diagnosis on the basis of the patient’s radiological and clinical features are required".
R: As requested, the lines are rewritten.
93) "information pertinent to the patient’s diagnosis may become" could be shortened to "patient’s diagnostic information becomes" (line 276).
R: As requested, the lines are rewritten.
94) "High risk group should be carefully evaluated clinical and radiological and should be screened with genetic test and by HRCT to prompt recognize early sign of ILD" (line 281) is not grammatically correct with respect to "evaluated clinical and radiological and should be screened".
R: As requested, the lines are rewritten.
95) "Updated diagnostic guidelines for IPF will hopefully reflect the growing knowledge on genetic component of the disease in a way that the clinical course of patients with specific clinical and radiological features and be able to provide innovative information on the molecular signatures predictive of UIP" (line 284) is too long and is not grammatically correct with respect to "clinical and radiological features and be able to". Please revise.
R: As requested, the lines are rewritten.
96) The meaning of "Thus, that accounts for inherited factors and HRCT patterns could be associated with different therapeutic approach" (line 287) is vague and the sentence is not grammatically correct with respect to "that accounts for inherited factors and HRCT patterns could be". Please rephrase this concluding statement to better highlight the role of genetic diagnosis in informing IPF therapy.
R: As requested, the lines are rewritten.
97) Please change "we described some association between" to "we described the association between" or "we have established a link between" (line 291).
R: As requested, the lines are rewritten.
98) Please replace "at time" with "at the time" (line 292).
R: As requested, the lines are rewritten.
99) "HRCT scans play a key role in IPF diagnosis and how clinical information, provided by pulmonologists and other clinicians, can aid radiologists in the interpretation of HRCT scans, guiding the need for surgical lung biopsy and determining available pharmacologic therapies" (line 293) is not grammatically correct with respect to "and how clinical information, provided by pulmonologists and other clinicians, can aid radiologists in the interpretation of HRCT scans".
R: As requested, the lines are rewritten.
100) "The central role of HRCT scans in the diagnosis of IPF may be enriched by genetic information to provide a differentiated therapeutic approach according to the molecular characteristics" (line 296) does not provide a straightforward and elegant conclusion. Please rephrase. In addition it is not clear what molecular characteristics are the authors referring to? Also, note that the word "role" was already used in the preceding sentence "HRCT scans play a key role in IPF diagnosis and how clinical information, provided by pulmonologists and other clinicians, can aid radiologists in the interpretation of HRCT scans, guiding the need for surgical lung biopsy and determining available pharmacologic therapies" (line 293).
R: As requested, the lines are rewritten.
101) Please distinguish the contribution of "Mario Santagiuliana" from that of "Michele Simbolo" in the Author Contributions section.
R: As required, the different authors’ roles have been specified
102) Please format "published version of the manuscript" (line 302) consistent with the rest of the text.
R: As requested, the lines are re-formatted.
103) Please change "funding" to "funding." (line 303).
R: As requested, the full-stop are added.
104) Please replace "Helsinki" with "Helsinki." (line 305).
R: As requested, the full-stop are added.
105) Please change "interest" to "interest." (line 306).
R: As requested, the full-stop are added.

Reviewer 2 Report
This is a very interesting and detailed paper describing the link between the radiographic appearance of IPF in patients with familial or sporadic IPF and any genetic variations therein. The study is limited by a small number of participants but nonetheless provides convincing evidence of a difference between the two groups studied.
Minor points.
Line 26-27. Please choose to use mean or average.
Lines 64-66, 102-105, 138-139,240-242. Please reformat.
Line 186-188. Please refer to figures rather than graphs.
The graphs would be improved by scaling the Y-axis from 0 to 100%. The Y-axis values do not need to be labelled with decimal places. Please create a caption labelled Figure X instead of Graph X, with a title and add the key here (remove from graph).
Author Response
This is a very interesting and detailed paper describing the link between the radiographic appearance of IPF in patients with familial or sporadic IPF and any genetic variations therein. The study is limited by a small number of participants but nonetheless provides convincing evidence of a difference between the two groups studied.
Minor points.
1) Line 26-27. Please choose to use mean or average.
R: We would like to thank the reviewer for these comments that have enabled us to enhance our manuscript.
2) Lines 64-66, 102-105, 138-139,240-242. Please reformat.
R: As requested, the lines have been reformatted.
3) Line 186-188. Please refer to figures rather than graphs.
R: As requested, the figures have been cited.
4) The graphs would be improved by scaling the Y-axis from 0 to 100%. The Y-axis values do not need to be labelled with decimal places. Please create a caption labelled Figure X instead of Graph X, with a title and add the key here (remove from graph).
R: We thank the reviewer for this suggestion. We rescaled Y-axis from 0 to 100%. Moreover, in the revised version, the Graph 1 and Graph 2 are labelled as Figure 2 and Figure 3 with the key lecture.
